

# Comparison of 1-year mortality predictions from vendor-supplied *versus* academic model for cancer patients

Michael F. Gensheimer[1], Jonathan Lu[2] and Kavitha Ramchandran[2]

[1] Department of Radiation Oncology, Stanford University School of Medicine, Stanford, CA, United States
[2] Department of Medicine, Stanford University School of Medicine, Stanford, CA, United States

## ABSTRACT

**Purpose:** The Epic End of Life Care Index (EOLCI) predicts 1-year mortality for a general adult population using medical record data. It is deployed at various medical centers, but we are not aware of an independent validation. We evaluated its performance for predicting 1-year mortality in patients with metastatic cancer, comparing it against an academic machine learning model designed for cancer patients. We focused on this patient population because of their high short-term mortality risk and because we had access to the comparator model predictions.

**Materials and Methods:** This retrospective analysis included adult outpatients with metastatic cancer from four outpatient sites. Performance metrics included AUC for 1-year mortality and positive predictive value of high-risk score.

**Results:** There were 1,399 patients included. Median age at first EOLCI prediction was 67 and 55% were female. A total of 1,283 patients were evaluable for 1-year mortality; of these, 297 (23%) died within 1 year. AUC for 1-year mortality for EOLCI and academic model was 0.73 (95% CI [0.70–0.76]) and 0.82 (95% CI [0.80–0.85]), respectively. Positive predictive value was 0.38 and 0.65, respectively.

**Conclusion:** The EOLCI's discrimination performance was lower than the vendor-stated value (AUC of 0.86) and the academic model's performance. Vendor-supplied machine learning models should be independently validated, particularly in specialized patient populations, to ensure accuracy and reliability.

## INTRODUCTION

Electronic medical record (EMR) vendor-supplied machine learning (ML) models aiming to predict mortality, clinical deterioration, or other outcomes have recently been deployed at many hospitals around the world (*Watson et al., 2020*; *Epic Systems, 2020a*; *Wong et al., 2021*; *Byrd et al., 2023*). Though the frequency of deployment and specific uses are usually not publicly available, some models are in widespread use: for instance, Epic Systems' Clinical Deterioration Index is being used at hundreds of hospitals (*Epic Systems, 2020a*). While these models have the potential to improve care and processes, there are also pitfalls as performance can drop from the advertised values, and use of ML models can have unintended consequences or worsen bias (*Zhang et al., 2022*; *Obermeyer et al., 2019*;

Corresponding author
Michael F. Gensheimer,
mgens@stanford.edu

*Gervasi et al., 2022*). It is important to validate models' performance in real-world usage since it can drop due to data distribution shift from the training data, coding errors, and other causes (*Zhang et al., 2022*; *Sahiner et al., 2023*; *Lu et al., 2022*). High-profile studies have shown lower than expected performance level of vendor-supplied models to predict sepsis or clinical deterioration in hospitalized patients (*Wong et al., 2021*; *Byrd et al., 2023*). Regulators have shown an interest in these issues; the United States federal government recently implemented new rules requiring increased transparency from EMR vendors' ML models (*Ross, 2023*).

While models focused on hospitalized patients have received more attention, vendors have also implemented models aimed at outpatients. The Epic End of Life Care Index (EOLCI) is a provider-facing model that predicts the risk of a patient dying in the next year. The goal is to identify patients at high risk of short-term mortality so teams can prioritize patients for advance care planning (*Epic Systems, 2022*). The EOLCI is deployed at our medical center and an unknown number of other centers, and can be added by users as a column to their patient lists. Accuracy of provider-facing models like the EOLCI is important to maintain trust and prevent alert fatigue which could result in truly high-risk patients not being contacted for advance care planning (*Shah et al., 2019*; *McGinn, 2016*). Through use with our own patients in oncology clinics, we noted that some of the EOLCI predictions did not seem very accurate. This is concerning since patients with metastatic cancer have high mortality and often a gradual decline in their health over the several years prior to death, so it is important that advance care planning is timely and accurate for this population (*Xu et al., 2022*; *Bestvina & Polite, 2017*).

We recently performed an evaluation of several mortality prediction models, including the EOLCI, comparing predictions to those of clinicians (*Lu et al., 2022*). In 150 hospitalized patients with cancer, we evaluated the EOLCI using an oncologist's answer to the 1-year surprise question as the surrogate outcome. For that setting, the EOLCI had an area under the receiver operating characteristic curve (AUC) of 0.7, sensitivity of 0.28, and positive predictive value (PPV) of 0.88. Thus, relative to clinicians, the model underestimated the risk level of these hospitalized cancer patients. This raises concern that this vendor-supplied model may not have high performance for the cancer patient population, though this study was limited by only using a clinician-defined surrogate outcome.

We are not aware of an external validation of the EOLCI for actual mortality outcomes, which motivated the current study in the outpatient oncology population at an academic medical center. This study is one of the first to evaluate a vender-supplied mortality model. We focused on patients with cancer for several reasons. First, as mentioned above, mortality prediction can be useful for these patients to help choose patients for interventions like advance care planning. Second, we had access to follow-up/death data for a large set of patients with cancer. Third, we were able to benchmark the EOLCI predictions against those from an internally developed cancer-specific survival model which has been in use in production since 2020, has been prospectively validated, and has shown value in helping to increase advance care planning conversations (*Gensheimer et al., 2019*, *2021*, *2024*). The findings may be of interest to sites with a need for mortality

predictions in a specialty population, which are considering options of deploying a vendor-supplied model like the EOLCI, training and using a home-grown model, or not using a predictive model at all.

## MATERIALS AND METHODS

### Epic end of life care index

The Epic EOLCI uses a logistic regression model on structured EMR data to predict the patient's risk of 1-year mortality. It was trained on and is designed for use in all adult patients. Model inputs include demographic, lab, comorbidity and medication information. A detailed model brief is available to Epic customers but is not publicly available (*Epic Systems, 2020b*, *2022*). The model was trained on routinely collected EMR data. Death/follow-up data to train the EOLCI was supplied by the clinical sites and it is unknown if high-quality death data sources like the National Death Index were used. The model output score ranges from 0 to 100 with higher score indicating higher mortality risk. Scores of 0–14 are considered low-risk, 15–45 medium-risk, and 45–100 high risk. The vendor stated that the lower threshold for the medium-risk group was chosen such that the sensitivity for 1-year mortality of the medium- to high-risk group was around 50%. The high-risk threshold was set to produce a positive predictive value of 25% for the high-risk group. The vendor reported AUC of 0.89–0.90 for test data from three sites. For cancer patients, the AUC was 0.86. The model brief mentions that the model may not be as well suited to cancer patients as others, because cancer stage is not reliably available as structured data in the EMR so cancer stage is not included as a predictor variable.

The model brief mentions that the score is not meant to be a calibrated prediction of mortality risk, but that the high risk group (score ≥45) had a PPV of 25%. The prevalence of 1-year mortality was around 1.5% in the population used for model development.

### Stanford model

We compared the EOLCI to our internally developed model trained in 2020, referred to as the Stanford model. It is a discrete-time survival model for cancer patients deployed for use in patient care since 2020 (*Gensheimer et al., 2019*, *2021*). An earlier version of the model was shown to have approximately doctor-level performance in predicting 1-year mortality (*Gensheimer et al., 2021*).

The Stanford model was trained in October 2020 with EMR data from 2008 to 2020 from two hospitals and several outpatient sites. Patients with metastatic cancer as identified by diagnosis codes, cancer registry, or cancer staging module in the EMR were included. Only patients with at least one note, lab result, and procedure code were included, to avoid including patients with no care at Stanford. A total of 15,410 patients were included, with an 80%/20% train/test split on the patient level.

For patients included in the Stanford cancer registry (the majority), follow-up/death data from the California Cancer Registry were used. Due to the inclusion of National Death Index data in the California Cancer Registry, nearly all deaths were captured (*Eisenstein et al., 2020*). For patients not in the cancer registry, EMR and Social Security Administration data were used for follow-up/death information.

The Stanford model has 4,279 predictor variables, including phrases from the text of provider notes and radiology reports, laboratory values, vital signs, and diagnosis and procedure codes. For notes, labs, and vital signs, the most recent one year of data was included; for the other data sources, all past data was included. More recent data was weighted more heavily; for details of the weighting, see a prior article (*Gensheimer et al., 2021*). For continuous predictor variables like vital signs and lab values, missing values were replaced with the mean value in the training set.

The model is a discrete-time survival model; it predicts the survival probability for each of 12 time intervals from 0 to 5 years of follow-up time. Figure 1 shows the model architecture. It was trained using Keras and the nnet-survival framework (*Gensheimer & Narasimhan, 2019*). L2 regularization was used to avoid overfitting; the strength was chosen using cross-validation to maximize log likelihood.

For this study we used only the predicted 1-year survival probability from the Stanford model. The prevalence of 1-year mortality in the test set used in development was 84.8%. The C-index for the test set of 3,082 patients used for model development was 0.78; AUC at specific follow-up time points was not calculated.

## Patient population

We included adult patients with metastatic cancer who had a return patient visit in a medical oncology clinic at four outpatient sites within Stanford Health Care (Emeryville, Palo Alto, Redwood City, and South Bay) during 2021–2022. These patients were chosen because they had prospective mortality predictions from the Stanford model that were made as part of a quality improvement initiative to increase advance care planning (*Gensheimer et al., 2023*). Once a week, the Stanford model predictions were generated for patients scheduled in clinic for the upcoming week and automatically saved in a CSV file. Presence of metastatic cancer was determined with the Epic cancer staging module and ICD-10 diagnosis codes.

We had access to EOLCI predictions for all patients in Stanford's Epic instance from four dates spaced throughout 2021–2022. If a patient had an EOLCI prediction on multiple dates, we only included the earliest eligible date. Patients who had a Stanford model prediction within the week prior to an EOLCI prediction were included in this analysis. We did not have Stanford model predictions at the exact EOLCI prediction dates, since the Stanford model predictions were made once a week. Thus, for each patient at a given EOLCI prediction date, their Stanford model prediction could be from 0 to 6 days in the past. We used the EOLCI prediction date as the starting date for model performance analysis, which would cause a slight bias in favor of the EOLCI since it had slightly more up-to-date information than the Stanford model.

## Statistical analysis

Model performance was assessed using Harrell's C-index, AUC for survival at various time points, Kaplan–Meier curves for quartiles of predicted mortality, and PPV for model-designated high risk patients. For performance analyses at a specific follow-up time point like AUC, patients lost to follow-up prior to that time point were excluded since we

**Input:**    4279-dimensional feature vector for a patient
**Output:**    Survival probability for each of 12 time intervals ranging from 0-5 years

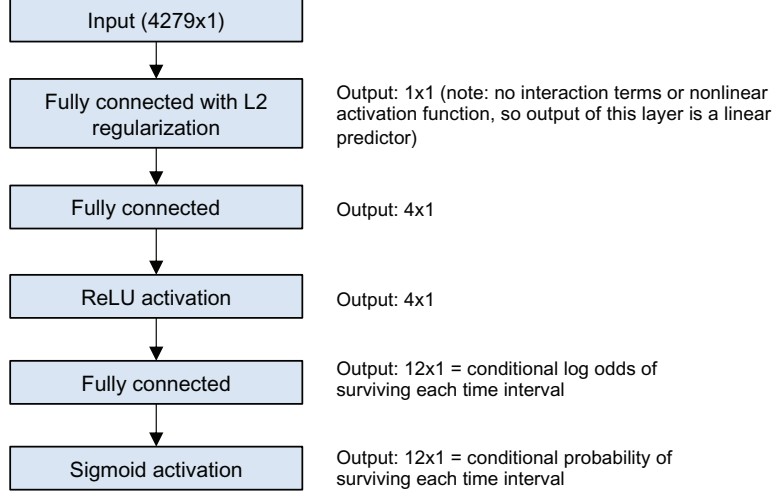

**Figure 1 Stanford model architecture.**

did not know if they were alive or dead at the time point of interest. There are pros/cons of various ways of handling such patients (*Reps et al., 2021*). Date of death was obtained from the EMR and the Social Security Administration Limited Access Death Master File. Details of the linkage to the Social Security Administration data were previously published (*Peralta, Desai & Datta, 2022*). For patients not known to be deceased, date of last follow-up was determined by finding the last encounter in the EMR. Death/follow-up data were obtained on February 25, 2024.

To help understand why EOLCI and Stanford model predictions differed, we wished to learn which variables were most influential for the Stanford model's predictions. For this, we used the following procedure, which relied on the fact that the output of the first layer of the neural network is a single value, a "linear predictor" which is a linear function of the predictor variables and coefficients. In a preprocessing step before making predictions, each predictor variable was standardized so that it had mean of 0 and standard deviation of 1 in the training set. For each patient, to find whether a predictor variable's value led to a longer or shorter predicted survival, we multiplied its value by its coefficient for the first layer of the neural network. To find the predictor variable $j$ increasing survival the most for patient $i$ compared to a "typical" patient with the mean value of $j$ in the training set, the following equation was used (*Gensheimer et al., 2021*):

$$\underset{j}{argmax} \; \beta_j \cdot x_{ij}$$

where $i$ is the index of the patient, $j \in \left\{1, \, 2, \, \ldots, \, n_{features}\right\}$ is the index of the predictor variable, $\beta$ is the vector of coefficients of the first layer of the neural network, and $x_{ij}$ is the value of predictor variable $j$ for patient $i$. The variables decreasing survival the most for each patient were found in an analogous way.

**Table 1** Patient characteristics (*n* = 1,399).

| Group | No. (%) or median (interquartile range) | |
| --- | --- | --- |
| | **All patients (*n* = 1,399)** | **Patients evaluable for 1-year mortality (*n* = 1,283)** |
| Age | 67 (56, 75) | 67 (56, 75) |
| **Sex** | | |
| Female | 770 (55%) | 706 (55%) |
| Male | 629 (45%) | 577 (45%) |
| Clinic (all medical oncology) | | |
| Palo Alto Breast | 161 (12%) | 145 (11%) |
| Emeryville* | 18 (1%) | 18 (1%) |
| South Bay* | 379 (27%) | 343 (27%) |
| Palo Alto Cutaneous | 59 (4%) | 53 (4%) |
| Palo Alto Gastroenterological | 5 (0.4%) | 5 (0.4%) |
| Palo Alto Gynecologic | 104 (7%) | 97 (8%) |
| Palo Alto Head and neck | 100 (7%) | 95 (7%) |
| Redwood City* | 73 (5%) | 63 (5%) |
| Palo Alto Sarcoma | 72 (5%) | 69 (5%) |
| Palo Alto Thoracic | 257 (18%) | 239 (19%) |
| Palo Alto Urologic | 171 (12%) | 156 (12%) |
| **Race** | | |
| Asian | 363 (26%) | 335 (26%) |
| Black | 38 (3%) | 32 (3%) |
| Other/multiple | 241 (17%) | 216 (17%) |
| Unknown | 25 (2%) | 21 (2%) |
| White | 732 (52%) | 679 (53%) |
| **Ethnicity** | | |
| Hispanic | 183 (13%) | 163 (13%) |
| Not Hispanic | 1,191 (85%) | 1,099 (86%) |
| Unknown | 25 (2%) | 21 (2%) |
| Epic End of life care index score (0–100) | 32 (12, 61) | 31 (12, 59) |
| Stanford model-predicted 1-year mortality probability | 0.22 (0.08, 0.46) | 0.21 (0.08, 0.45) |

**Note:**
 * At these outreach clinics, patients with all cancer types were seen.

This retrospective study was approved by the Stanford University Institutional Review Board with waiver of consent (#47101). R version 4.2.3 was used for analysis. The article complies with the TRIPOD+AI reporting guidelines (*Collins et al., 2024*).

# RESULTS

## Patient characteristics

There were 1,399 patients included. Patient characteristics are in Table 1. Median age at first EOLCI prediction was 67 and 55% were female. Median follow-up was 16.3 months (range 0.03–33). A total of 488 patients (35%) were deceased. Median survival was not reached. By the Kaplan–Meier method, 1- and 2-year survival was 78% (95% CI [76–80])

**Table 2 Discrimination metrics.**

| Timepoint | Num. evaluable patients | Metric | Epic EOLCI | Stanford model |
|---|---|---|---|---|
| – | 1,399 | C-index | 0.69 [0.67–0.72] | 0.77 [0.75–0.78] |
| 180 days | 1,342 | AUC | 0.74 [0.71–0.78] | 0.85 [0.82–0.88] |
| 1 year | 1,283 | AUC | 0.73 [0.70–0.76] | 0.82 [0.80–0.85] |
| | | Positive predictive value | 0.38 | 0.65 |
| | | Sensitivity | 0.63 | 0.75 |
| | | Specificity | 0.70 | 0.73 |
| | | Positive likelihood ratio | 2.07 | 2.79 |
| | | Negative likelihood ratio | 0.54 | 0.34 |
| | | Diagnostic odds ratio | 3.87 | 8.10 |

**Note:**
95% confidence intervals in parentheses. For area under the ROC curve, only patients evaluable for that timepoint were included (deceased, or had follow-up past that timepoint).

and 64% [61–67], respectively. A total of 1,283 patients were deceased or had >1 year follow-up and so were evaluable for the 1-year mortality endpoint. Of these, 297 (23%) died within 1 year.

## Model performance

Performance metrics are shown in Table 2 and ROC curves are in Fig. 2. AUC for 1-year mortality was 0.73 (95% CI [0.70–0.76]) for the EOLCI and 0.82 [0.80–0.85] for the Stanford model. Results for 180-day mortality were similar. AUC results for demographic subgroups are in Table 3; the Stanford model had better performance than the EOCLI for all subgroups. Kaplan–Meier curves for tertiles of model-predicted mortality are in Fig. 3; there was greater separation between the three groups for the Stanford model.

Of the 1,283 patients evaluable for the 1-year mortality endpoint, 484 had high risk of mortality according to the EOLCI (score 45 or greater). Of EOLCI high-risk patients, 186 (38%) died within 1 year; of EOLCI low-risk patients, 111 (14%) died within 1 year. Thus, for EOLCI high-risk patients, the positive predictive value for 1-year mortality was 0.38, which is higher than the 0.25 reported by Epic in a general patient population. The Stanford model did not have a pre-specified high-risk score cutoff, so we created one by choosing a score cutoff such that 484 patients would be flagged as high-risk (to match the number of EOLCI high-risk patients); this cutoff was >31% predicted probability of 1-year mortality. At this cutoff, the Stanford model's positive predictive value for high-risk patients was 0.65. The diagnostic odds ratio for the EOCLI was 3.87 and for the Stanford model it was 8.10. This measure is the ratio of the odds of the patient being predicted to die within 1 year if they truly died within 1 year, relative to the odds of the patient being predicted to die within 1 year if they survived >1 year. We also calculated the Gini index for each model; this is reported in the Supplemental Results section (*Wu & Lee, 2014*).

For metrics at a specific follow-up time point such as AUC and positive predictive value, patients lost to follow-up prior to the follow-up time point of interest were excluded. As a sensitivity analysis, we re-calculated these performance metrics without excluding such patients but instead counting them as alive (reported in Supplemental Results section). The

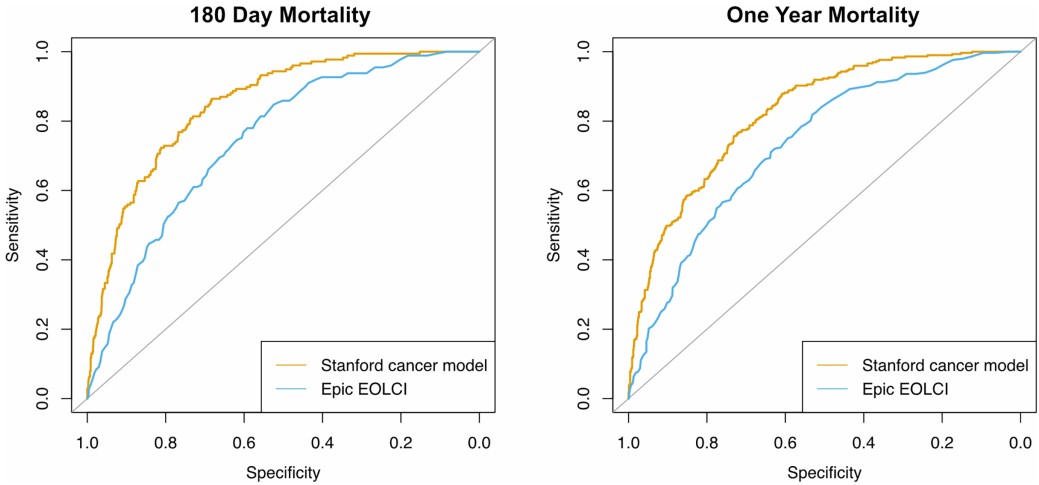

**Figure 2** Receiver operating characteristic curves for prediction of mortality at 180 days (*n* = 1,342) and 1 year (*n* = 1,283). Only patients evaluable for that timepoint were included.

**Table 3 AUC for 1-year mortality in demographic subgroups (in 1,283 evaluable patients).**

|  | No. patients | Epic EOLCI AUC | Stanford model AUC |
|---|---|---|---|
| **Age tertile** | | | |
| 22–60 | 428 | 0.81 [0.76–0.86] | 0.85 [0.81–0.89] |
| 61–72 | 427 | 0.72 [0.66–0.78] | 0.80 [0.76–0.85] |
| 73–99 | 428 | 0.70 [0.65–0.75] | 0.82 [0.77–0.86] |
| **Sex** | | | |
| Female | 706 | 0.73 [0.69–0.78] | 0.83 [0.80–0.87] |
| Male | 577 | 0.72 [0.67–0.76] | 0.81 [0.77–0.85] |
| **Race** | | | |
| Asian | 335 | 0.73 [0.66–0.79] | 0.84 [0.79–0.88] |
| Black | 32 | 0.75 [0.57–0.94] | 0.85 [0.71–0.99] |
| Other/multiple | 216 | 0.70 [0.62–0.79] | 0.78 [0.71–0.85] |
| Unknown | 21 | 0.42 [0.03–0.81] | 0.85 [0.63–1.0] |
| White | 679 | 0.75 [0.71–0.79] | 0.83 [0.79–0.86] |
| **Ethnicity** | | | |
| Hispanic | 163 | 0.71 [0.62–0.80] | 0.79 [0.72–0.87] |
| Not Hispanic | 1,099 | 0.74 [0.71–0.77] | 0.83 [0.80–0.85] |
| Unknown | 21 | 0.55 [0.18–0.55] | 0.70 [0.50–0.90] |

**Note:**
95% confidence intervals in parentheses.

Stanford model performance was still superior to the EOLCI, though its positive predictive value was lower than the main analysis at 0.40.

To get insight into why the Stanford model's performance was higher in this evaluation, for each patient we recorded the Stanford model's five predictor variables that had the most positive influence on predicted survival time, and the five variables with the most

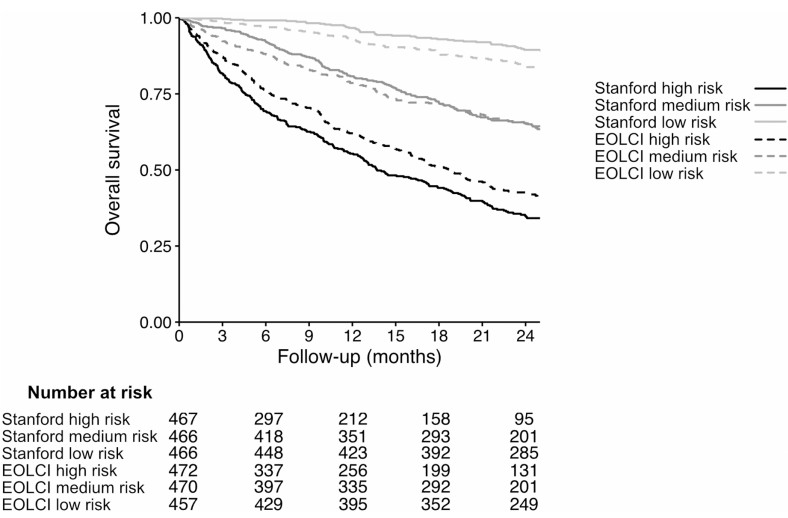

**Figure 3 Kaplan–Meier curves for patients in tertiles of model-predicted mortality, for Stanford model (solid lines) and EOLCI model (dashed lines).** The range of predicted 1-year mortality for the Stanford high-, medium-, and low-risk tertiles was 37.4–86.6%, 12.1–37.4%, and 0.04–12.1%, respectively. The range of EOLCI scores for the EOLCI high-, medium-, and low-risk tertiles was 52–99, 19–51, and 0–18, respectively.

negative influence. The most common of these high-influence variables are in Table 4. The EOLCI's predictor variables are not publicly available so we are not able to include a full comparison with them. Some high-influence variables in the Stanford model are also predictors for the EOLCI, such as age and albumin, and some are not, such as denosumab and granulocyte count.

## DISCUSSION

The performance of the EOLCI was lower for our patient population than previously reported. AUC of the EOLCI for 1-year survival was 0.73, much lower than the value of 0.86 for cancer patients reported by the vendor. The Stanford cancer-specific model had better performance with AUC of 0.82. The PPV of the EOLCI for 1-year mortality was higher than that reported by the vendor for a general population, but this was expected since our population had a much higher 1-year mortality rate than the EOLCI training population (23% *versus* 1.5%). It is not surprising that the Stanford model had higher observed performance than the EOLCI, since it has many more predictor variables, was trained specifically on patients with metastatic cancer, and was trained on Stanford patients who may differ from the patients from other centers used to train the EOLCI. Also, the cut-point for the EOLCI high-risk group was not tuned for a metastatic cancer population with high short-term mortality rate, which could affect the EOLCI's PPV in this study. Since the EOLCI predictor variables are not publicly available, it is difficult to do a comprehensive analysis of the differences between the EOLCI and Stanford model variables, but several of the most influential Stanford model variables such as albumin and pulse have been validated as mortality predictors in other studies (*Gupta & Lis, 2010*; *Gann et al., 1995*; *Lee et al., 2016*). Easier interpretability is an advantage of the EOLCI compared to models like the Stanford one that have many more parameters. Many of the Stanford

**Table 4 Stanford model predictor variables with largest influence on predicted survival for 1,399 patients.** For each patient, the five variables with the largest positive influence on predicted survival, and the five with the largest negative influence, were recorded. The most common variables found in these lists are shown.

| Variable | Explanation | # patients |
| --- | --- | --- |
| **Variables Improving Predicted Survival** | | |
| High albumin level | – | 469 |
| Low pulse | – | 190 |
| Low age | – | 173 |
| Note term: "adult inpatient" | Used in the phrase "adult inpatient plan of care" in notes by nurses and physical therapists | 127 |
| High "albumin ser/plas" level | | 93 |
| Note term: "nrbc abs" | Lab result: absolute number of nucleated red blood cells in blood | 80 |
| Note term: "granulocyte abs" | Lab result: absolute number of granulocytes in blood | 73 |
| Diagnosis: V62.89 (other psychological or physical stress not elsewhere classified) | – | 63 |
| Note term: "the inpatient" | Usually refers to inpatient services such as "the inpatient orthopedic service" | 60 |
| Note term: "c[10–19]" | Simplified term produced by the note preprocessing. Found when the letter C is followed by any number from 10 to 19. Usually when the note includes ICD-10 diagnosis codes such as C18.9 (colon cancer). | 59 |
| **Variables Worsening Predicted Survival** | | |
| Note term: "total by" | Sometimes used to indicate total sum of tumor diameters for RECIST measurements. | 806 |
| Note term: "mg total" | Used in phrases like "up to 200 mg total" | 419 |
| Procedure: 1123F (advance care planning) | – | 406 |
| High age | – | 288 |
| High pulse | – | 197 |
| Medication: denosumab | – | 126 |
| High anion gap | – | 116 |
| Medication: electrolytes | – | 84 |
| Diagnosis: V66.7 (encounter for palliative care) | – | 80 |
| Medication: morphine | – | 79 |

model predictor variables are collinear and related to each other in complex causal pathways which should lead to caution in interpreting their coefficients. Others do not have straightforward interpretations, such as the note text term "total by" which predicted shorter survival. This term was commonly used in notes tracking the size of cancer masses. These notes are mainly used for patients on clinical trials for metastatic cancer, which are often used after standard therapies have failed, so would be correlated with shorter survival time. Explanations for model predictions are not necessary for all applications; for instance in the implementation of the Stanford model there are no explanations given to users, but the deployment has still been successful (*Gensheimer et al., 2024*).

The goal of comparing the EOLCI to the Stanford model was not to learn which would work better for another health system's patients or a general patient population, since the Stanford model was trained using Stanford-specific data such as custom procedure codes and would not work well at another hospital or for patients without cancer. Instead, the

Stanford model serves as an example of an internally developed model for a specific use case. Hospitals must sometimes decide between using a vendor-supplied model or developing their own. This study may help decision-makers understand that (1) the performance of a vendor-supplied model will likely be lower than the values reported in the vendor's initial evaluation, and (2) training their own model could offer significant performance gains.

There are several potential reasons for the decreased performance of the EOLCI compared to the vendor-stated values. The relationship between predictor variables and outcomes could be different from the training data; for instance, the EOLCI relies on hand-entered diagnosis codes in the EMR and coding practices vary between institutions (*Sahiner et al., 2023*). Also, the model was not trained specifically to achieve high performance in a metastatic cancer population, and this may be a challenging population for performance metrics since there is uniformly a high mortality risk among these patients.

Any model's performance needs to be evaluated in the context of the intended use (*Li, Asch & Shah, 2020*; *Shah, Milstein & Bagley, 2019*). In conjunction with our previous study finding that the EOLCI underestimated risk level for hospitalized cancer patients compared to a physician, the current findings shed light on suitable uses for the EOLCI (*Lu et al., 2022*). The EOLCI had modest performance in the current study, with AUC for 1-year mortality of 0.73 and diagnostic odds ratio of 3.87. Its performance may be borderline for consequential decision-making or higher-risk populations. In such a setting, its use may cause providers to experience frustration and alert fatigue from low performance level, resulting in users not performing the model-suggested interventions (*McGinn, 2016*; *Buchan et al., 2020*). Development of population- and site-specific AI models such as the Stanford model may help address these issues, though such models come with their own challenges such as the need for a team with specialized and varied expertise to create them, and support from the hospital information technology department for integration into the EMR and monitoring of data quality and performance (*Zhang et al., 2022*; *Sahiner et al., 2023*). In addition to assessing models' predictive performance, it is important to measure the clinical impact of their use. There are currently limited data on the clinical use of the EOLCI, though one abstract reported that it was useful for triaging inpatients for palliative care referral and led to shorter time from hospital admission to consult (*Grandhige, 2024*).

There are several limitations to this study. One is the quality of survival data used for evaluation of the models' performance. Several studies have shown that EMR data can under-capture deaths and lead to overestimation of survival time due to informative censoring (sicker patients being more likely to be lost to follow-up, causing bias) (*Curtis et al., 2018*; *Gensheimer et al., 2022*). To help address this limitation, we supplemented the EMR data with Social Security Administration death data, but these data have also been shown to be incomplete (*Levin et al., 2019*). This incomplete data issue especially affects calibration metrics, since in the presence of informative censoring a well-calibrated model will appear to over-predict mortality risk. Discrimination metrics like AUC may be less affected by incomplete data (*Raclin et al., 2023*). Another limitation is that we tested the

EOLCI's performance in a specific patient population, and performance could be higher in a general patient population such as in a primary care clinic. Finally, we only included patients from a single institution and results could be different at other centers.

## CONCLUSIONS

The EOLCI's performance in this group of patients with metastatic cancer was lower than that reported during model development. Along with other recent studies reporting worse-than-expected performance of vendor-supplied ML models, the results highlight the importance of validating such models in the patient population in which they will be deployed.

## ACKNOWLEDGEMENTS

We thank Daniel Rubin, Balasubramanian Narasimhan, Solomon Henry, Douglas Wood, and Sigi Javitz for curating and supplying data used to train the Stanford model.

### Funding

Supported by the National Cancer Institute (Cancer Center Support Grant number 5P30CA124435) and National Institutes of Health/National Center for Research Resources (CTSA award number UL1 RR025744). Jonathan Lu was supported by the Stanford Baxter Foundation Graduate Student Scholar Award, Stanford University School of Medicine MedScholars grant, and the Stanford Medicine Program for AI in Healthcare, which is funded by a gift from Debra and Mark Leslie as well as the Department of Medicine and Stanford Healthcare. The funders had no role in study design, data collection and analysis, decision to publish, or preparation of the manuscript.

### Grant Disclosures

The following grant information was disclosed by the authors:
National Cancer Institute: 5P30CA124435.
National Institutes of Health/National Center for Research Resources: UL1 RR025744.
Stanford University School of Medicine MedScholars.
Department of Medicine and Stanford Healthcare.

### Competing Interests

The authors declare that they have no competing interests.

### Author Contributions

- Michael F. Gensheimer conceived and designed the experiments, performed the experiments, analyzed the data, prepared figures and/or tables, authored or reviewed drafts of the article, and approved the final draft.
- Jonathan Lu conceived and designed the experiments, authored or reviewed drafts of the article, and approved the final draft.

- Kavitha Ramchandran conceived and designed the experiments, authored or reviewed drafts of the article, and approved the final draft.

## Human Ethics

The following information was supplied relating to ethical approvals (*i.e.*, approving body and any reference numbers):

The Stanford University IRB granted approval to carry out this retrospective study.

## Data Availability

The anonymized patient dataset is available in the Supplemental File.

The code for the current analysis and a list of predictor variables for the Stanford model is available at GitHub:

https://github.com/MGensheimer/prognosis-model/tree/master/epic_versus_stanford_models.

The code for training the Stanford model is available at GitHub:

https://github.com/MGensheimer/prognosis-model/tree/master/radiation_oncology_survey_2020/code.

## Supplemental Information

Supplemental information for this article can be found online at http://dx.doi.org/10.7717/peerj.18958#supplemental-information.

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
