# Peer review of "Comparison of 1-year mortality predictions from vendor-supplied versus academic model for cancer patients"

_PeerJ, doi:10.7717/peerj.18958_

## Round 0.1 · original submission · Major Revisions

After carefully considering the reviewers' comments, I recommend major revisions to address several important issues raised. The reviewers have highlighted the potential value of your study but also noted areas requiring clarification and further analysis to ensure the rigor and applicability of your findings.

Key points for revision include:

-Enhancing the clarity and consistency of your methods and reporting, particularly around key metrics such as PPV and model thresholds.
-Providing more detailed justifications for methodological choices (e.g., exclusion criteria for follow-up data) and addressing ambiguities in the presentation of your results.
-Expanding the discussion to include interpretations of feature importance and broader context regarding the practical implementation and comparative utility of the models analyzed.
-Revisiting and refining certain aspects of data presentation (e.g., Table 1 format) for consistency and reader comprehension.

Both reviewers also suggest strengthening the introduction and discussion by situating your findings more explicitly in the broader context of predictive model development and implementation.

Please submit a revised manuscript addressing these points, along with a detailed response to reviewers outlining how each comment has been addressed.

We look forward to your revised submission.

·

Basic reporting

a. Line 101: “Scores of 0-14 are considered low-risk, 15-45 medium-risk, and 45-100 high risk.”  I was unclear what medium-risk meant to Epic, consider adding (from the Epic Overview document) their definitions: medium-risk = sensitivity set at 50%; high-risk = PPV set at 25%.
b. Lines 162-163: “If a patient had an EOLCI prediction on multiple dates, we only included the earliest eligible date."  Consider moving this sentence to the second sentence in that same paragraph for better readability, as I was assuming until then that you were using multiple prediction instances per patient for your analysis which would have been unsound.
c. Lines 167-168: “For performance analyses at a specific follow-up time point like AUC, patients lost to follow-up prior to that time point were excluded.”  Please explain why you chose to exclude them, when in theory you still had their death/no-death outcome in your EMR and SSA Death Master File repository? Additionally the phrase “like AUC” is ambiguous – please explain if you excluded these patients for other metrics in addition to AUC.
d. Table 1, last row “Stanford model-predicted 1-year mortality risk”  You have these values as %, which is inconsistent with the rest of the table and % should not have an IQR. Technically I see that the % changes your probability (0.22) to a percent to compare with the EOLCI value, but seeing the % sign here associated with an IQR is confusing, and my opinion would be to remove the % and report the actually probability (ie “0.22 (8,46)”) for both groups.

Experimental design

a. I appreciate the authors’ emphasis on PPV, as this is arguably the most clinically relevant metric for clinicians in the context of these models. However, I am concerned with the approach taken to create comparable PPV values between the EOLCI and the Stanford model, as described in lines 216 to 220. Specifically, although you adjusted the Stanford model threshold to flag the same number of high-risk patients as EOLCI at a score of 45, such a threshold adjustment could obscure other performance aspects, such as sensitivity, specificity, or the overall calibration of probabilities. I strongly recommend that you explicitly report the associated sensitivities of both models at the thresholds you chose (the same thresholds that you got your PPV values from).

Validity of the findings

a. The authors spend a great deal of time describing their method for determining feature importance and display the most important features in Table 4. I was hoping for more interpretation of these results in your discussion. My takeaway from the discussion section (lines 240-243) is that albumin and pulse were important in the Stanford model, and other models have shown that they are important. But what about the #1 most important variable for predicting worsening survival: “Note term: ‘total by’”? I appreciate that NLP tokens meanings can be mysterious, but I would greatly appreciate it if the authors could speculate or show examples in the literature of why they think their model is placing such importance on this term (and on the other non-intuitive ones, like “Note term: ‘adult inpatient’”). Epic’s EOLCI variables are all intuitive and discrete and have a straightforward ‘importance’ viewing feature in the EHR - one could argue that this interpretability outweighs the performance gains of a potentially less interpretable model such as the Stanford Model. However, based on your successful implementation of this model (reference #21), I suspect you could successfully counter-argue this point (see next comment).
b. In the Discussion section, the authors state that “Any model’s performance needs to be evaluated in the context of the intended use” (line 263). Although the focus of this paper is the external validation of EOLCI, as opposed to the Stanford model (which is the topic of several existing papers references throughout the manuscript), I believe it would be very helpful for the reader to state here briefly or in the introduction section that the Stanford model has actually been implemented and solving the problem it was intended to. This would ground the comparison between the two models in reality. Then, are there any scholarly descriptions of other systems that have tried or are using EOLCI? A quick google search shows this abstract (https://www.sciencedirect.com/science/article/pii/S088539242400366X) which shows positive results from EOLCI implementation in the hospital setting when paired with palliative care consults. Perhaps the EOLCI can be helpful to a general patient population (granted this is one small study), but in the outpatient setting, for metastatic cancer patients specifically, it remains unproven and may perform worse than advertised (as you mention).
c. Line 270: “… though such models come with their own challenges.”  Such as? If intending to inform healthcare systems on the pros and cons of off-the-shelf vs. custom models, some insight into exactly what issues they could expect with a custom model would be useful.
d. Lines 277-278: “Mitigating this issue, even incomplete survival data can be successfully used to validate model discrimination performance”.  This sentence confuses me – you state earlier in the paragraph that several studies have shown that EMR data can lead to overestimation of survival time, yet here you state that incomplete survival data can be used to validate model performance. Please clarify.

Additional comments

The manuscript provides a comprehensive comparison of the EOLCI and Stanford predictive models for predicting 1-year mortality among patients with metastatic cancer. The EOLCI is understudied and this paper has the potential to shed light on its performance among this unique patient population. Addressing the issues highlighted above would enhance the clarity, rigor, and applicability of the findings, ensuring that readers can fully appreciate the nuances and implications of this important work.

Reviewer 2 ·

Basic reporting

This paper compares 1-year mortality predictions for cancer patients using vendor-supplied models versus academic models. It is an interesting and valuable study that was a pleasure to read. My comments are as follows.
1. The positive predictive value (PPV) of a prediction model heavily depends on the population to which the model is applied. As the authors acknowledged in lines 107–108 of the paper, “The model brief mentions that the score is not meant to be a calibrated prediction of mortality,…”, they should therefore avoid being overly critical of the vendor-supplied model for underestimating the mortality risk of very high-risk cancer patients in this study.
2. From lines 212 to 220, I calculated the following metrics for the vendor-supplied model: a positive likelihood ratio of (186/297) / (298/986) = 2.07, a negative likelihood ratio of (111/297) / (688/986) = 0.54, and a diagnostic odds ratio of 2.07 / 0.54 = 3.87. While the prediction performance of the vendor-supplied model is not perfect and is inferior to the academic model, it still deserves some recognition, especially considering that the academic model (Stanford model) was specifically developed for this population.
3. From lines 212 to 220, I also calculated the Gini index (equivalently the Pietra index for a prediction model that produces a binary output of high-risk and low-risk predictions) for the vendor-supplied model. The calculation is as follows: the index is given by multiplying the proportion of high-risk predictions by the proportion of low-risk predictions, then multiplying by the sum of the positive predictive value (PPV) and negative predictive value (NPV) minus one, and dividing by the one-year mortality rate and its complement. Using the data provided, the calculated Gini index is 0.32. This means the vendor-supplied model achieves 32% of the performance of a perfect prediction model that predicts all patients to die or survive within one year without fail. This is worth noting, as it underscores the model’s relative utility. (Reference: Alternative performance measures for prediction models. PLoS ONE2014;9:e91249.)

Experimental design

This paper compares 1-year mortality predictions for cancer patients using vendor-supplied models versus academic models. It is an interesting and valuable study that was a pleasure to read. My comments are as follows.
1. The positive predictive value (PPV) of a prediction model heavily depends on the population to which the model is applied. As the authors acknowledged in lines 107–108 of the paper, “The model brief mentions that the score is not meant to be a calibrated prediction of mortality,…”, they should therefore avoid being overly critical of the vendor-supplied model for underestimating the mortality risk of very high-risk cancer patients in this study.
2. From lines 212 to 220, I calculated the following metrics for the vendor-supplied model: a positive likelihood ratio of (186/297) / (298/986) = 2.07, a negative likelihood ratio of (111/297) / (688/986) = 0.54, and a diagnostic odds ratio of 2.07 / 0.54 = 3.87. While the prediction performance of the vendor-supplied model is not perfect and is inferior to the academic model, it still deserves some recognition, especially considering that the academic model (Stanford model) was specifically developed for this population.
3. From lines 212 to 220, I also calculated the Gini index (equivalently the Pietra index for a prediction model that produces a binary output of high-risk and low-risk predictions) for the vendor-supplied model. The calculation is as follows: the index is given by multiplying the proportion of high-risk predictions by the proportion of low-risk predictions, then multiplying by the sum of the positive predictive value (PPV) and negative predictive value (NPV) minus one, and dividing by the one-year mortality rate and its complement. Using the data provided, the calculated Gini index is 0.32. This means the vendor-supplied model achieves 32% of the performance of a perfect prediction model that predicts all patients to die or survive within one year without fail. This is worth noting, as it underscores the model’s relative utility. (Reference: Alternative performance measures for prediction models. PLoS ONE2014;9:e91249.)

Validity of the findings

This paper compares 1-year mortality predictions for cancer patients using vendor-supplied models versus academic models. It is an interesting and valuable study that was a pleasure to read. My comments are as follows.
1. The positive predictive value (PPV) of a prediction model heavily depends on the population to which the model is applied. As the authors acknowledged in lines 107–108 of the paper, “The model brief mentions that the score is not meant to be a calibrated prediction of mortality,…”, they should therefore avoid being overly critical of the vendor-supplied model for underestimating the mortality risk of very high-risk cancer patients in this study.
2. From lines 212 to 220, I calculated the following metrics for the vendor-supplied model: a positive likelihood ratio of (186/297) / (298/986) = 2.07, a negative likelihood ratio of (111/297) / (688/986) = 0.54, and a diagnostic odds ratio of 2.07 / 0.54 = 3.87. While the prediction performance of the vendor-supplied model is not perfect and is inferior to the academic model, it still deserves some recognition, especially considering that the academic model (Stanford model) was specifically developed for this population.
3. From lines 212 to 220, I also calculated the Gini index (equivalently the Pietra index for a prediction model that produces a binary output of high-risk and low-risk predictions) for the vendor-supplied model. The calculation is as follows: the index is given by multiplying the proportion of high-risk predictions by the proportion of low-risk predictions, then multiplying by the sum of the positive predictive value (PPV) and negative predictive value (NPV) minus one, and dividing by the one-year mortality rate and its complement. Using the data provided, the calculated Gini index is 0.32. This means the vendor-supplied model achieves 32% of the performance of a perfect prediction model that predicts all patients to die or survive within one year without fail. This is worth noting, as it underscores the model’s relative utility. (Reference: Alternative performance measures for prediction models. PLoS ONE2014;9:e91249.)

Additional comments

This paper compares 1-year mortality predictions for cancer patients using vendor-supplied models versus academic models. It is an interesting and valuable study that was a pleasure to read. My comments are as follows.
1. The positive predictive value (PPV) of a prediction model heavily depends on the population to which the model is applied. As the authors acknowledged in lines 107–108 of the paper, “The model brief mentions that the score is not meant to be a calibrated prediction of mortality,…”, they should therefore avoid being overly critical of the vendor-supplied model for underestimating the mortality risk of very high-risk cancer patients in this study.
2. From lines 212 to 220, I calculated the following metrics for the vendor-supplied model: a positive likelihood ratio of (186/297) / (298/986) = 2.07, a negative likelihood ratio of (111/297) / (688/986) = 0.54, and a diagnostic odds ratio of 2.07 / 0.54 = 3.87. While the prediction performance of the vendor-supplied model is not perfect and is inferior to the academic model, it still deserves some recognition, especially considering that the academic model (Stanford model) was specifically developed for this population.
3. From lines 212 to 220, I also calculated the Gini index (equivalently the Pietra index for a prediction model that produces a binary output of high-risk and low-risk predictions) for the vendor-supplied model. The calculation is as follows: the index is given by multiplying the proportion of high-risk predictions by the proportion of low-risk predictions, then multiplying by the sum of the positive predictive value (PPV) and negative predictive value (NPV) minus one, and dividing by the one-year mortality rate and its complement. Using the data provided, the calculated Gini index is 0.32. This means the vendor-supplied model achieves 32% of the performance of a perfect prediction model that predicts all patients to die or survive within one year without fail. This is worth noting, as it underscores the model’s relative utility. (Reference: Alternative performance measures for prediction models. PLoS ONE2014;9:e91249.)

---

## Round 0.2 · accepted · Accept

The authors addressed the reviewers' concerns and substantially improved the content of the manuscript. So, based on my assessment as an academic editor, the manuscript can be accepted in its current form.

·

Basic reporting

All of my critiques have been thoroughly addressed.

Experimental design

All of my critiques have been thoroughly addressed. I appreciate the additional sensitivity analysis including all patients including those lost to follow-up and the results are informative and well interpreted.

Validity of the findings

All of my critiques have been thoroughly addressed. The enhanced description of the Stanford model variable meanings adds validity to the study.

Additional comments

The authors have clearly taken substantial time and effort to address all critiques. I highly commend their efforts and revisions, and I strongly recommend this paper for publication.

Reviewer 2 ·

Basic reporting

The authors have adequately addressed my comments in revising their study.

Experimental design

The authors have adequately addressed my comments in revising their study.

Validity of the findings

The authors have adequately addressed my comments in revising their study.

Additional comments

The authors have adequately addressed my comments in revising their study.